# Prototype of Nitro Compound Vapor and Trace Detector Based on a Capacitive MIS Sensor

**DOI:** 10.3390/s20051514

**Published:** 2020-03-10

**Authors:** Nikolay Samotaev, Artur Litvinov, Maya Etrekova, Konstantin Oblov, Dmitrii Filipchuk, Alexey Mikhailov

**Affiliations:** 1National Research Nuclear University MEPhI (Moscow Engineering Physics Institute), Kashirskoe hwy 31, 115409 Moscow, RussiaMOEtrekova@mephi.ru (M.E.);; 2“INKRAM” LLC Research and Production Company (“Inkram” RPC LLC), Mikhalkovskaya Street 63 “Б”, Bldg. 1, Floor 3, Premise VII, 125438 Moscow, Russia; mikhailov@inkram.ru

**Keywords:** field effect gas sensor, MIS structure type Pd–SnO_2_–SiO_2_–Si, NO_2_ gas detection in ppb level, pyrolysis of nitro compound vapor

## Abstract

A prototype of a nitro compound vapor and trace detector, which uses the pyrolysis method and a capacitive gas sensor based on the metal–insulator–semiconductor (MIS) structure type Pd–SiO_2_–Si, was developed and manufactured. It was experimentally established that the detection limit of trinitrotoluene trace for the detector prototype is 1 × 10^−9^ g, which corresponds to concentration from 10^−11^ g/cm^3^ to 10^−12^ g/cm^3^. The prototype had a response time of no more than 30 s. The possibility of further improving the characteristics of the prototype detector by reducing the overall dimensions and increasing the sensitivity of the MIS sensors is shown.

## 1. Introduction

Modern problems of industrial and civil safety, environmental monitoring and health care are solved by a wide range of technical means, including devices for gas analysis. An urgent task is to control the concentration of gases NO_2_, NH_3_, H_2_S, C_2_H_5_S and Cl_2_, which are also likely decomposition products of dangerous and harmful chemicals. For example, for organic nitro compounds that are part of most explosives, one of the products of thermal decomposition is NO_2_ [1,2,3].

Currently, to solve the problem of the operational detection of explosives including nitroaromatics at critical infrastructure facilities (transport, industrial enterprises, cultural events) portable detectors based on the method of ion-mobility spectrometry (IMS) are often used. The IMS instruments have high sensitivity in the ppb range, like gas chromatography (GC), but also have the ability to operate at atmospheric pressure, a fast response in the order of seconds, and a relatively simple design [4,5,6]. However, such devices are expensive and not so easy to operate.

The limit of detection (LOD) of trinitrotoluene (TNT) for commercially available IMS detectors is 10^−14^ g/cm^3^ within the operating range of mass detected substances is from 1.0 × 10^−11^ g to 2.0 × 10^−7^ g based on vendor information. 

Less sensitive, yet simpler and cheaper methods, which are based on the effect of changing the luminescence intensity of a sensor based on organic photoactive chemosensory compositions in the presence of nitro compounds, are also known [7,8,9,10]. However, there are also significant disadvantages of fluorescent quenching (FQ) methods: sensor materials’ instability and complicated synthesis, intermolecular self-quenching and photo-bleaching.

The LOD of TNT for commercially available FQ detectors is from 10^−8^ g/cm^3^ to 10^−12^ g/cm^3^ for vapor and 1.0 × 10^−9^ g for trace amounts based on vendor information.

The practice of using the mentioned devices also has revealed their common significant drawback, which is the impossibility of conducting covert searches due to the sufficiently large overall dimensions without the prospect of significant miniaturization.

Therefore, the relevance of scientific research remains and there are a lot of methods for detecting explosives, many of which do not even have prototypes today [11,12,13].

Stand-off usage methods with non-contact sampling, like Raman spectroscopy, laser-induced breakdown spectroscopy, Terahertz spectroscopy and so on, are of unconditional interest. 

However, despite the promising idea, these methods are bulky, expensive and have a number of issues, the most restrictive is low sensitivity. According to [14,15], the LOD of the high energetic materials for Raman-based apparatus is attained with the trace surface density from 0.5 μg/cm^2^ to 100 μg/cm^2^ in the case of a remote detection range of 5–10 m. This density, rated by authors on the order of several ppm, means from 10^−7^ g/cm^3^ to 10^−8^ g/cm^3^ in terms of TNT.

Another promising area of research is the wide variety of gas sensors: chemical sensors [16,17,18,19], sensors and miniature systems fabricated by using Complementary Metal Oxide Semiconductor (CMOS) technology [20,21,22,23,24], micro and nanoelectromechanical sensors and systems (MEMS and NEMS) including mechanical sensors, such as cantilever beams or gravimetric sensors [20,25,26] and others.

The LOD of the high energetic materials (or NO_2_ as a product of pyrolysis) for the mentioned gas sensors averages from hundreds of ppt for cantilever beams to hundreds of ppb for semiconductor and chemical sensors, with response times at the level of several minutes.

From the point of view of miniaturization issues, two leading groups of gas sensors are distinguished [27]: (1) carbon nanotube (CNT) and ZnO nanowire gas sensors, (2) semiconductor- and polymer-based gas sensors. 

The gravimetric sensors, CNTs and other nanomaterial gas sensors are very promising, but these technologies are only a recent development and there are still fundamental challenges that need to be addressed before they can be commercialized sensors. Nevertheless, CNT technology helps improve existing microelectronic gas sensors [21]. 

Semiconductor gas sensors include gas-sensitive devices based on field effect or work function changes [28]: metal oxide semiconductor-based (MOS), field effect transistor-based (FET), MOSFET-based and MIS sensors based on successive film structures of metal, insulator and semiconductor.

The methods for increasing the sensitivity and selectivity of microelectronic gas sensors by using pre-concentrators [29] and pattern recognition algorithms in conjunction with a multiple-sensor array [30,31] are also known. However, for the efficient operation of the matrix of gas sensors included in a machine learning neural network, long-term stability of characteristics and high selectivity are required, which, to this day, are urgent tasks for research in the field of gas sensors.

In this paper, we will talk about a highly gas-sensitive (at the ppb-level) capacitive sensor based on a MIS structure with a palladium electrode made using pulsed laser spraying technology [32]. The first references to gas sensors with a similar design and operating principle are found in [33]. 

The principle of operation of gas capacitive MIS sensors [34] is as follows. Molecules of gases from the atmosphere are adsorbed on the surface of the metal (palladium) electrode and diffuse through the porous film of Pd-gate to the metal-insulator (MI) interface. The MI surface is a transition region of variable stoichiometric composition, where gas molecules are adsorbed on “traps” [35]. As a result of adsorption, a change in the electron density occurs in the transition region of the MI, which leads to a redistribution of charge carriers in the semiconductor near the insulator-semiconductor (IS) interface and a change in the total capacitance of the MIS structure. A change in the capacitance of the MIS structure is registered by the electronic circuit as a signal of the MIS sensor.

In previously published works [36,37], the conditions for the detection of vapors and traces of nitro compounds (using the example of TNT) using the pyrolysis method and an MIS sensor with a Pd–SiO_2_–Si structure (diameter of the Pd-gate d_Pd_ = 3 mm) are detailed, which recorded the concentration of gaseous products of the thermal decomposition of nitro compounds. 

It was experimentally established [37] that the optimal mode of operation of the MIS sensor when responding to TNT vapor decomposition products corresponds to holding a gaseous sample in a pyrolysis chamber (reactor) at a temperature of 500–550 °C for 1 s. The result obtained is in good agreement with the theoretical foundations of the thermal decomposition of TNT [3].

The optimal operating parameters of the experimental setup were determined and it was shown that the proposed method has a detection limit of TNT in vapor 1 × 10^−12^ g/cm^3^ and in particles 1 ng.

The aim of this stage of work was the design, assembly, and testing of the first-of-its-kind prototype of a nitro-containing substance detector based on a microelectronic sensor device using the effects of charge changes in a capacitive MIS structure.

## 2. Design and Manufacture of Prototype Nitro Compounds Detector

At the design stage of the detector prototype, several significant drawbacks of the experimental setup used in [37] were taken into account and corrected:The analyzed volume of the gas path was reduced from 40 to 14 cm^3^;The length and power consumption of the pyrolysis chamber (reactor, Figure 1) were reduced from 25 to 13 cm, from 92 to 45 W, respectively;The design of the evaporator has been optimized, namely a constructive solution with replaceable sample preparation modules of two types has been used: for operation in the vapor registration mode (heated PTFE-tube) and in the registration mode of traces (toaster-type design, Figure 2).

In accordance with a separate study [38], it was decided that implementing an electronic circuit for measuring the sensor’s capacitance by the amplitude bridge method within the prototype of the detector would provide the most sensitive measurements. The sensitivity of the sensor when applying a concentration of *C*_NO2_ = 110 ppb is 14 rel.un./ppm. The noise level of the electronic circuit with an MIS sensor is ±0.005 rel.un. Therefore, taking the value of the triple amplitude of noise as the minimum value of the useful signal, we have the calculated LOD of NO_2_ concentration, equal to
(1)$${\mathrm{LOD}}_{{\mathrm{NO}}_{2}}=\frac{3\times 0.005}{14}\frac{\text{rel.un.}}{\text{rel.un./ppm}}=1.1\;\mathrm{ppb}.$$

Therefore, from the calculation that each TNT molecule will thermally decompose to form one NO_2_ molecule, one can expect a minimum detectable concentration of TNT vapors at the level of 10^−11^ g/cm^3^.

The draft and photograph of the measuring unit of the prototype detector are shown in Figure 3.

The principle of operation of the prototype detector is as follows. Sampling of the air containing vapors of the analysis substance is carried out by a diaphragm pump. A napkin made of aluminum foil with trace amounts of TNT deposited on its surface is laid in the evaporator. The process of TNT sample preparation is described in [37]. 

TNT vapors obtained by sublimation in an evaporator at a temperature of 80 °C enter a reactor where, under the influence of a temperature of 450 °C, gas-phase thermal decomposition of TNT molecules occurs with the formation of NO_2_, the concentration of which is detected by the MIS sensor. As in earlier experiments, the type of structure of the sensor is Pd–SiO_2_–Si, the diameter of the electrode is *d*_Pd_ = 3 mm, and the operating temperature is T_MIS_ = 100 °C.

The signal from the MIS sensor is processed by an electronic analog board, digitized by analog-to-digital converter (ADC) and fed to a control unit based on a microcontroller, which also controls the light and sound alarms.

The rotameter regulator is used for the possibility of regulation and visualization of the flow rate of the gas mixture (from 0.1 dm^3^/min to 1 dm^3^/min). To maintain a stable temperature for heating the reactor and the sample preparation modules, thermoregulators are used, the operation of which is an implemented using solid-state relay. The design of the reactor and evaporator provides temperature sensors. Power sources, temperature regulators and relays are placed in a separate unit, connected via a cable to the measurement unit. The prototype of the detector is powered by 220 V.

Studies [39] showed that the sensitivity of MIS sensors to water vapor can be the cause of false alarms when recording low gas concentrations. Therefore, to compensate for the influence of the humidity on the readings of the MIS sensor, a temperature and humidity sensor is provided as part of the detector prototype, the readings of which are used in real time to calculate the compensation correction:
*N*(*P*) = *N*_0_ + *A*(1 − exp(−*γP*)),

where *P* is the vapor pressure of water, torr; *N*_0_, *A*, and *γ* are the coefficients determined by measuring the dependence of the MIS sensor readings on the pressure of water vapor (Figure 4). The result of moisture compensation is a modified MIS sensor signal, hereafter referred to as MIS_Signal_Compensated.

As you can see, with a smooth change in water vapor pressure (not more than 10 torr in 1–3 min, which, at an ambient temperature, for example, 27 °C, corresponds to a change in relative humidity by 20%), the compensation of the MIS sensor readings for humidity works as efficiently as possible. With more drastic changes in humidity (50% or more), deviations of the MIS_Signal_Compensated value from the zero level is ±0.5 rel.un. (means ±36 ppb of NO_2_), due to the difference in response rates because of a rapid change in the concentration of water vapor for the humidity sensor and directly for the MIS sensor. 

## 3. Nitro Compounds Detector Prototype Test Results

After powering up the prototype detector, the stabilization of the MIS sensor drift at a constant level not exceeding ±(10–20) rel.un./min on average is 30–60 min. However, if digital processing is applied to the initial signal of the MIS sensor, to calculate the first-order derivative d(MIS_Signal)/*dt*, then the exit time to the operating mode is reduced to 10–15 min.

Using signal differentiation, it also made it possible to reduce the response time by almost four times (dt = 10 s) and reduce the recovery time, as will be shown below.

The response of the detector prototype to TNT samples was tested in two modes with the corresponding sample preparation modules for analysis of TNT trace (Figure 2, toaster-type) and vapor (Figure 3, tube-type). However, it is worth noting that both sample preparation modules acted as an evaporator and the difference between the modes was only in the size and shape of the aluminum foil, which served as a substrate for the trace sample (Figure 5).

Figure 6 shows the results of testing the prototype detector for TNT samples: the MIS_Signal_Compensated is in the upper part of the graphs and the differential signal is in the lower part (*dt* = 10 s).

As you can see, the sample preparation module tube-type is more efficient than a toaster-type evaporator, which can be explained by the imperfection of the design second (large dilution of the vapor sample TNT with air). However, the advantage of the toaster-type evaporator is the more stable zero level of the MIS sensor.

Thus, it was experimentally confirmed that, for our nitro compound prototype detector based on an MIS sensor with a Pd–SiO_2_–Si structure (*d*_Pd_ = 3 mm), the detection limit of TNT is 1 ng. According to the estimate, assuming that the TNT traces are completely evaporated during the entire duration of its supply (3 min) at a sampling flow of 0.25 dm^3^/min, 1 ng corresponds to a concentration of 1.3 × 10^−12^ g/cm^3^.

The reason for the difference between the result obtained and the previously calculated level of 10^−11^ g/cm^3^ may be as follows. The sensitivity of MIS sensors to NO_2_ is nonlinear and increases with decreasing gas concentration [37]. Besides, decreasing sampling flow from 0.50 to 0.25 dm^3^/min caused the pyrolysis mode to change and each TNT molecule become the source of not one, but several NO_2_ molecules.

According to the test results using the software of the detector prototype, the light and sound alarm threshold was set at the level of d(MIS_Signal)/*dt* = −0.03, which corresponds to the LOD of TNT at the level of units of ng and the vapor concentration at the level of 5 × 10^−12^ g/cm^3^.

Figure 6 also shows the possibility of tracking false alarms due to the reaction to reducing gases (H_2_, NH_3_, and H_2_S), using the example of the NH_3_ background concentration. Due to the evident difference in the response dynamics, the probability of false positives of the detector can be easily compensated by software.

As can be seen from Figure 7 (special cases of Figure 6b), the time the differential signal reaches the detection threshold of the detector prototype (dashed line) is 15–20 s for a TNT mass of 1 μg and 30 s for TNT mass 2 ng.

The recovery time of the differential signal readings is 3 min for a TNT mass of 1 μg and 1 min for TNT mass 2 ng.

The generalized results of tests of the prototype detector for samples of TNT of various weights are shown in Figure 8.

Therefore, it was shown that the LOD of TNT for the nitro compound prototype detector based on a capacitive MIS sensor (Pd–SiO_2_–Si structure, d_Pd_ = 3 mm) is from 10^−11^ g/cm^3^ to 10^−12^ g/cm^3^ for vapor and 1 × 10^−9^ g for trace. The response time is no more than 30 s. The time required for readiness for new measurements is no more than 5 min. Optimal settings: evaporator temperature 80 °C, reactor temperature 450 °C, sampling rate 0.3–0.5 dm^3^/min.

The result obtained in comparison with existing mass-produced portable detectors is very competitive. However, taking into account the prospects for further improvement of the developed detector (miniaturization and reduction of energy consumption), it is necessary to have a margin for the sensitivity of the sensor element.

## 4. Investigation of the Possibility of Increasing the Sensitivity of MIS Sensors

One of the possible ways to increase the sensitivity of the MIS sensor is to increase the area of the Pd-gate.

To establish the dependence of the sensitivity of MIS sensors to NO_2_ on the area of the Pd-gate, we studied sensors with the Pd–SiO_2_–Si structure, differing only in the diameters of the Pd-gate: 1; 2; 3 and 6 mm. The results are presented in Figure 9.

As can be seen, a 30-fold increase in the area of the Pd-gate gives an increase in sensitivity by about 10 times. Note that MIS sensors with a Pd–SiO_2_–Si structure and a Pd-gate diameter of 1 mm do not have sensitivity sufficient to solve the problem posed in this work, and sensors with a large gate diameter of 6 mm are subject to a significant continuous decrease in sensitivity in the first six months of operation.

MIS sensors with a diameter of a Pd-gate from 2 to 3 mm, which were used in the prototype of the detector, have the optimal operating parameters of sensitivity and stability of characteristics for recording gas concentrations at the ppb level, but are not miniature and energy efficient: dimensions of 1 cm^3^ (Figure 10a), power consumption of 1 W at T_MIS_ = 150 °C.

The miniaturization and reduction of energy consumption problem was solved using a specially developed technology for the manufacture of semiconductor sensor elements in the form factor of the SMD package [40,41,42]. A high-tech planar design of an MIS sensor made of ceramic materials was developed and created, which allows the minimization of heat loss due to the decrease in warm dispersion to the sensor housing (Figure 10b).

To solve the problem of increasing the sensitivity of the MIS sensor, a new type of MIS structure, Pd–SnO_2_–SiO_2_–Si, was fabricated and studied, characterized in that the dielectric layer of SnO_2_ was additionally sprayed onto the SiO_2_ film by the magnetron sputtering of tin in the presence of oxygen. This method allows the formation of a dielectric film with a highly effective surface area, which, according to [34], can contribute to a significant increase in sensitivity. The experimental results obtained on the electronic circuit for measuring the sensor’s signal using capacitance-to-digital converter [38] confirm this (Figure 11).

As can be seen, SnO_2_ sensors, in comparison with SiO_2_ sensors, have significantly higher sensitivity even with a smaller diameter of Pd-gate: Nine times more for T_MIS_ = 100 °C;Thirty-two times more for T_MIS_ = 140 °C.

A detailed comparison of the sensor characteristics is given in the Table 1.

A comparison of the characteristics of SnO_2_ and SiO_2_ sensors shows the possibility of achieving NO_2_ sensitivity an order of magnitude higher and to reduce LOD of NO_2_ by three times, even with a smaller Pd-gate diameter. It was found that the maximum NO_2_ sensitivity for SnO_2_ sensors corresponds to operation temperature T_MIS_ = 140 °C, which is consistent with the data of several other authors [23,43]. However, the reducing response time for SnO_2_ sensors is another significant target that is achieved at T_MIS_ = 170 °C.

## 5. Discussion and Conclusions

A prototype of a first-of-its-kind vapor detector of nitro compounds based on a microelectronic capacitive MIS sensor was developed and tested.

Taking into account the previously performed work [36,37], it was shown that the pyrolysis method with a detector based on the MIS sensor retains the ability to detect TNT at the level of masses 1 × 10^−9^ g and concentrations from 10^−11^ g/cm^3^ to 10^−12^ g/cm^3^, with a significant decrease in the size of the chamber pyrolysis.

It was found that a simple sensor signal processing algorithm allows one to achieve selectivity with respect to water vapor and background gases and reduce the reaction time by a factor of four, the response time τ_0.9_ no more than 30 s and the recovery time τ_0.1_ no more than 3 min. 

Further perspectives on miniaturization and improvement of the sensitivity characteristics of MIS sensors was demonstrated; for example, modifying the metal–insulator interface with a SnO_2_ layer and using specially developed flexible laser micromilling technology.

The important role of the sample preparation module (namely, the evaporator in this work) is shown experimentally which made it possible to illustrate the difference between the LOD for the method and the LOD for the detector. High-energy organic substances, including nitro compounds, have extremely low saturated vapor pressure at room temperature [16,44]. This makes it clear why the LOD for the detector, due to the inevitable loss during the sampling, is much higher than the LOD for the method. Therefore, it is so important to have a margin of sensitivity for the sensor element.

This method for detecting nitro compounds using pyrolysis and an MIS sensor does not claim to be the most sensitive, most selective, and most miniature solution. Nevertheless, the presented method allows for the achievement a more important result, from the point of view of the authors, of the optimal ratio of price and quality.

Thus, the pyrolysis method, using a gas capacitive MIS sensor as a sensitive element, can be successfully applied to create detectors of nitro compounds for contact (and, in the future, hidden) inspection in real time.

## Figures and Tables

**Figure 1 sensors-20-01514-f001:**
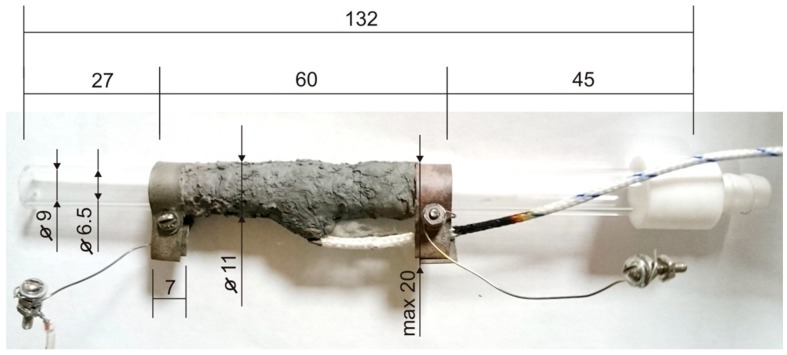
The pyrolysis chamber (reactor). Dimensions are given in mm.

**Figure 2 sensors-20-01514-f002:**
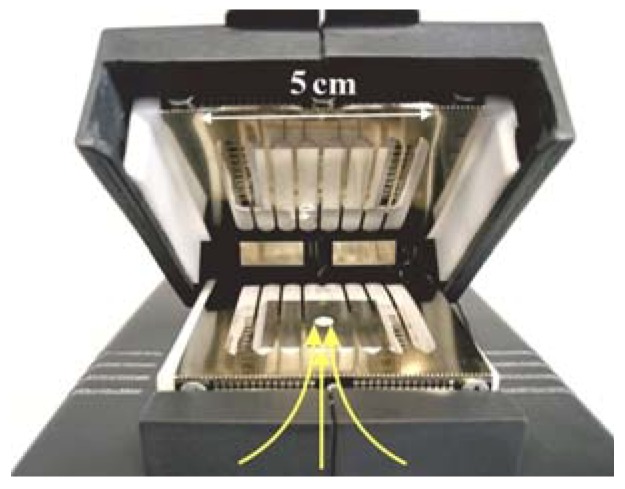
Sample preparation module toaster-type design.

**Figure 3 sensors-20-01514-f003:**
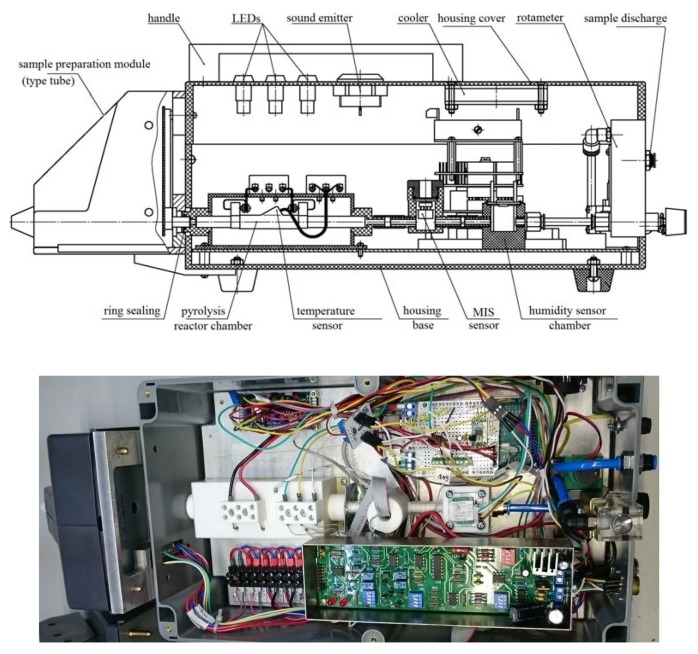
The draft and photograph of prototype detector of nitro compounds (measurement unit) with a sample preparation module for operation in the vapor registration mode. Overall dimensions of the measuring unit 530 × 210 × 200 mm (L × W × H).

**Figure 4 sensors-20-01514-f004:**
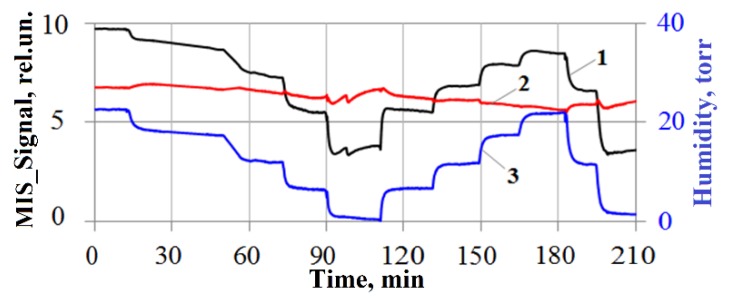
Compensation of the readings of the MIS sensor for humidity: curve (1) the initial MIS_Signal, (2) MIS_Signal_Compensated, (3) water vapor pressure.

**Figure 5 sensors-20-01514-f005:**
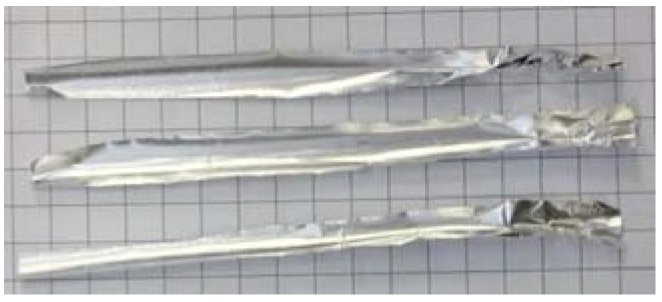
Appearance of TNT samples for working with a sample preparation module in the form of a heated tube.

**Figure 6 sensors-20-01514-f006:**
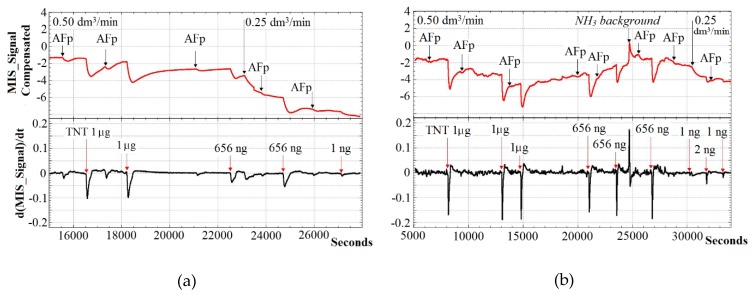
The response of the detector prototype to TNT samples. Working in trace analysis mode with a toaster-type sample preparation module (**a**). Working in the vapor analysis mode with a sample preparation module like a tube (**b**). Designation “AFp” means pure aluminum foil.

**Figure 7 sensors-20-01514-f007:**
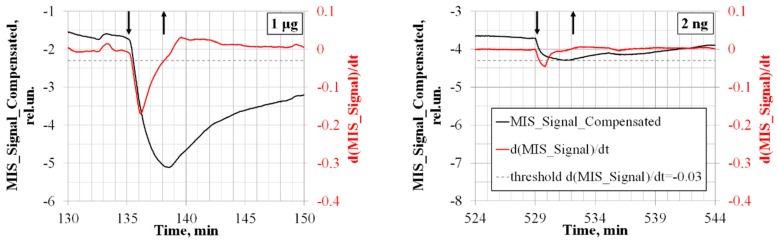
The time to reach the trinitrotoluene (TNT) detection threshold for samples with different masses and the dynamics of the recovery time detector prototype readings. The time of the sample submission is 3 min (shown by arrows).

**Figure 8 sensors-20-01514-f008:**
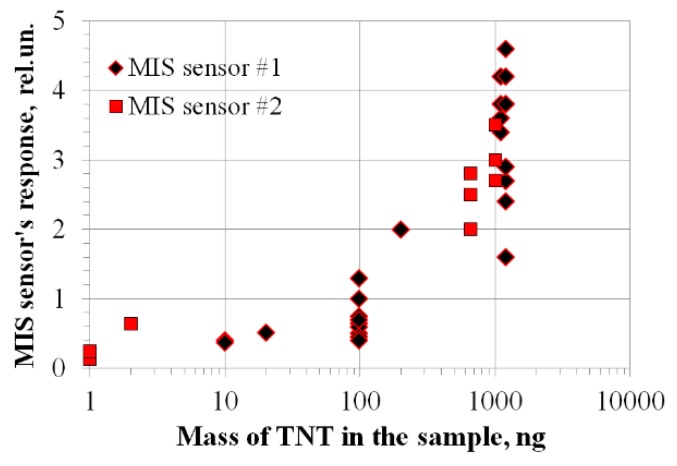
Calibration of a prototype detector using TNT samples of various weights.

**Figure 9 sensors-20-01514-f009:**
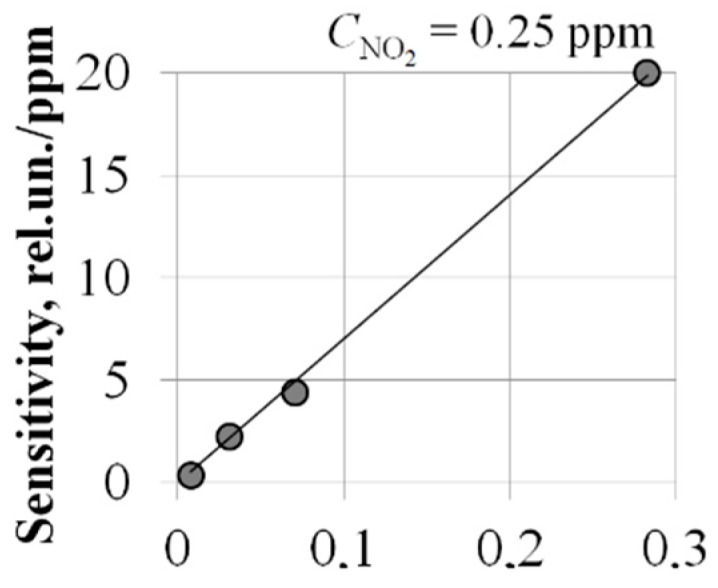
The dependence of the sensitivity of metal–insulator–semiconductor (MIS) sensors with the structure of Pd–SiO_2_–Si to NO_2_ on the area of the Pd-gate. T_MIS_ = 100 °C.

**Figure 10 sensors-20-01514-f010:**
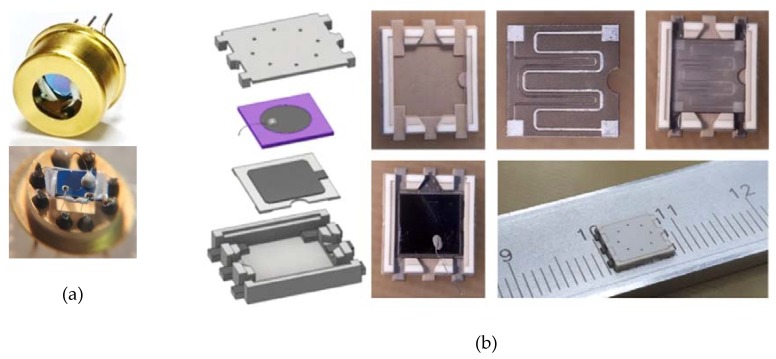
Scheme of the device MIS sensor in a typical sensor housing (**a**). An electronic model of the MIS sensor design components in a package of monolithic ceramics fabricated by the developed flexible laser micromilling technology (**b**).

**Figure 11 sensors-20-01514-f011:**
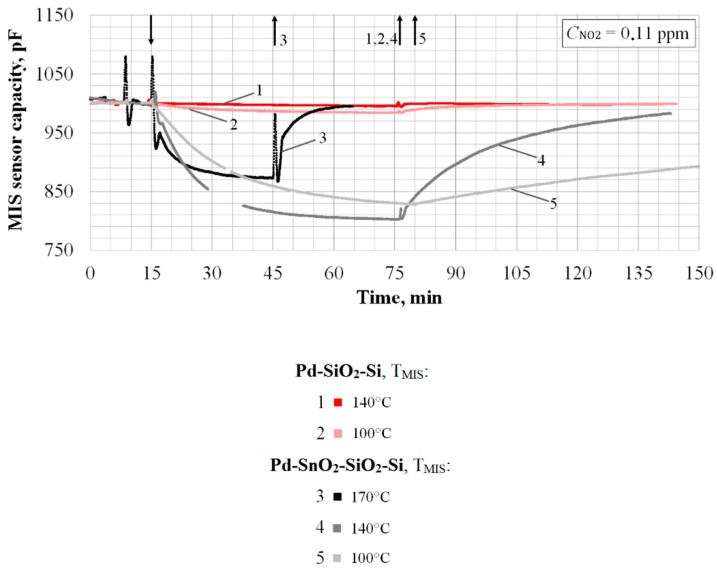
Responses MIS sensors with structures Pd–SiO_2_–Si (*d*_Pd_ = 3 mm) and Pd–SnO_2_–SiO_2_–Si (*d*_Pd_ = 2 mm) to concentration of NO_2_ 0.11 ppm at different sensor temperature T_MIS_. The moments of the beginning and finishing of the concentration submission are shown by arrows.

**Table 1 sensors-20-01514-t001:** The comparative characteristics of the capacitive MIS sensors with structures Pd–SiO_2_–Si and Pd–SnO_2_–SiO_2_–Si.

	Pd–SiO_2_–Si: *d*_Pd_ = 3 mm; Noise Level ±0.2 pF			
T_MIS_, °C	NO_2_ Sensitivity ^1^, pF/ppm	Response Time τ_0.9_, min	Recovery Time τ_0.1_, min	LOD ^2^ of NO_2_, ppb
100	175	40	60	3.4 ^3^
140	55	-	-	11.0
	**Pd**–**SnO_2_**–**SiO_2_**–**Si**: *d*_Pd_ = 2 mm; noise level ±0.5 pF			
100	1565	45	250	1.0
140	1775	25	65	0.9
170	1180	15	15	1.3

^1^ For case *C*_NO2_ = 0.11 ppm. ^2^ LOD is calculated by analogy with Equation (1). ^3^ Here, the LOD differs from the previously calculated 1.1 ppb due to different electronic circuit for measuring the sensor’s capacitance [38].

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
