# Peer review of "Prototype of Nitro Compound Vapor and Trace Detector Based on a Capacitive MIS Sensor"

_sensors, 2020, doi:10.3390/s20051514_

Round 1

Reviewer 1 Report

Reviewer: Minor revision

Comments for the paper “Prototype of nitro compounds vapor and trace detector based on a capacitive MIS sensor”. This work developed a capacitive gas sensor based on the MIS structure type Pd-SiO2-Si. The techniques used are adequate and the results are properly presented. The topic is interesting for the target readerships, but there are still some problems to be solved before possible publication. My specific comments are as follows:

Introduction: For highlighting MIS gas sensor, the research status of gas sensors and related gas sensors’ type is insufficient. In addition, the first five paragraphs do not cover any references, which do not respect academic norms, and it is necessary to quote relevant literature in the research background. Please refer to some literature about gas sensor research like: Sensors 2020, 20(2), 408; Sensors & Actuators: B. Chemical 298 (2019) 126874; J. Electron. Mater., 2017, 46, 6895-6900; Nano Energy, 2019, 58, 312-321. The full name of “MIS” in the abstract should be provided. “the” in page 2: Initial needs to be capitalized. “NO2” in page 2, “SnO2” in Figure 11: Number requires subscript. Please check the full manuscript. The sensing mechanism should be better explained in the manuscript and the relevant citation should be included. Figures quality needs to be improved, such as font size and linewidth. The language should be improved throughout the manuscript.

Author Response

Thank you for your detailed feedback on our work.

Your comments and additions were very valuable and helped us significantly expand and improve the manuscript.

Reviewer 2 Report

Authors present describe the overall performance of a nitro compounds vapor and trace sensor. According to authors the device is based on a microelectronic sensor device using the effects of charge changes in a capacitive MIS structure, which was able to detect TNT in vapor 1x10-12 g/cm3 and in particles 1 ng. Even when the equipment can be interesting I consider that the manuscript lacks in quality and also in novelty, since authors did not discuss the technical principles of any part of the equipment, just focus on the overall performance of the full equipment. Moreover, the manuscript is a kind of compendium of several articles published by authors, this issue can be clearly observed in the reference list, where practically all are self-citations. Here authors did not reviewed in literature looking for similar sensors or at least that detect the same gases in order to have reference points to compare their device. Hence I consider that the manuscript do not have enough quality for be published in Sensors and therefore I suggest to reject it.

Author Response

Thank you for your feedback.

Your comments and additions helped us significantly expand and improve the manuscript.

Reviewer 3 Report

In this work the author describe a prototype sensor device designed for monitoring nitro compounds, using a metal-insulator-semiconductor element (Pd-SiO2-Si) as sensing element.

In the first part of their work, the authors illustrate the technical improvements that they achieved in comparison with their previous results. Altough the nature of this part of the work is mainly incremental, it is technically sound and well presented in the most fundamental points. 

In the second part, the authors discuss about the margins for improvement in the sensitivity of the MIS sensing element, showing a significant improvement in the absolute response by adopting a Pd-SnO2-SiO2-Si structure. Even if the largest signal for Pd-SnO2-SiO2-Si is also accompanied by a slow recovery time (for lower working temperatures), a decent compromise seems to be obtained at 170°C working temperature.

There is clearly still room for improvement and, although the work covers a specific field of application, I think that the work can be of interest to the readers,

However, some minor issues can be evidenced: 

- The first paragraphs of the introduction (lines 1 to 40) have no reference at all. Authors should cite some published work to help the reader to understend better the topic. For example, some citations in line 34 regarding the cheaper methods using changes of luminescence intensity ahould be provided with some reference on the revelant numbers. That is, for example, authors might report the limit of detection of typical fluorescence-based sensors and compare them to the one obtained by the more expensive spectrometers based on on mobility.

- Line 31: Please introduce here the acronym TNT as it is here that the word trinitrotoluene is used for the first time.

- Line 31. What is the meaning of the three dots? Do you mean that 10^-11 to 10^-7 g is the typical range of the detection limit? Please also check the units. 1.0x10^(-11) has no units.

- Line 103: please define what you mean for MDP. Also there is a spelling error ("Compencated" instead of "compensated"). The same spelling error is in the caption of Figure 4 

- Line 108: same spelling error ("compencated")

- The overall discussion of figure 11 is not very clear. First, I assume that the NO2 is insereted at t=0 in the figure, but it is not clearly mentioned. Moreover, it look as the recovery for the 5 curves statrts at different points, i.e. at about 80 min for four sensors (curves 1,2,4 and 5) and at about 40 minutes for curve 30. If that is correct, the Authors should mention it too

Moreover, the discussion is no very clear. I refer to the lines 187.192:

"Ratios of sensitivity to NO2 at various operating temperatures of sensors TMIS = 100; 140 and 187 170 °C have the form: 188
“1 - 0.3 - no data” - for sensors with a Pd-SiO2-Si structure (maximum sensitivity is observed at 189 TMIS = 100 °C) and 190
"0.9 - 1 - 0.7" - for sensors with the Pd-SnO2-SiO2-Si structure (maximum sensitivity is observed 191 at TMIS = 140 °Ð¡)."

What author exactkly meant? I also guess that inserting the results in a Table would be much better for the sake of clarity.

Author Response

(The authors gave the same response as above.)

Reviewer 4 Report

Manuscript ID: sensors-722217

This submission may become suitable for publication in sensors if the following minor revisions are addressed appropriately.

Authors should briefly comment on the superiority of this MIS capacitive sensor compared to other related methods in the literature in terms of design of the reactor and efficiency of sensing. Literature updation is required since some key reviews are missing. Authors should revise the manuscript for English corrections and spell check.

Author Response

Thank you for your feedback.

Your comments helped us significantly expand and improve the manuscript.
